# Rethinking a Non-Predominant Pattern in Invasive Lung Adenocarcinoma: Prognostic Dissection Focusing on a High-Grade Pattern

**DOI:** 10.3390/cancers13112785

**Published:** 2021-06-04

**Authors:** Yeonu Choi, Jonghoon Kim, Hyunjin Park, Hong Kwan Kim, Jhingook Kim, Ji Yun Jeong, Joong Hyun Ahn, Ho Yun Lee

**Affiliations:** 1Department of Radiology and Center for Imaging Science, Samsung Medical Center, Sungkyunkwan University School of Medicine (SKKU-SOM), 81 Irwon-Ro, Gangnam-Gu, Seoul 06351, Korea; skyblue718@skku.edu; 2Department of Electrical and Computer Engineering, Sungkyunkwan University, Suwon 16419, Korea; jhkim4915@skku.edu (J.K.); hyunjinp@skku.edu (H.P.); 3Center for Neuroscience Imaging Research, Institute for Basic Science, Suwon 16419, Korea; 4Department of Thoracic and Cardiovascular Surgery, Samsung Medical Center, Sungkyunkwan University School of Medicine (SKKU-SOM), Seoul 06351, Korea; hkts@skku.edu (H.K.K.); jkimsmc@skku.edu (J.K.); 5Department of Pathology, Kyungpook National University Chilgok Hospital, Kyungpook National University School of Medicine, Daegu 41404, Korea; jyjeong@knu.ac.kr; 6Biomedical Statistics Center, Data Science Research Institute, Samsung Medical Center, Sungkyunkwan University School of Medicine (SKKU-SOM), Seoul 06351, Korea; anter@skku.edu

**Keywords:** lung adenocarcinoma (ADC), heterogeneity, high-grade pattern, histology, prognosis, recurrence

## Abstract

**Simple Summary:**

Prognostic considerations for non-predominant histologic patterns are necessary because most lung adenocarcinomas have a mixed histologic pattern. We aimed to identify prognostic stratification by second most predominant pattern of lung adenocarcinomas and to more accurately assess prognostic factors with CT imaging analysis, particularly enhancing non-predominant but high-grade pattern. We confirmed that the second most predominant histologic pattern can stratify lung adenocarcinoma patients according to prognosis. Especially, when the second most predominant pattern was high-grade, recurrence risk increased by 4.2-fold compared with the low-grade group. Thus, predicting the malignant potential and establishing treatment policies should not rely only on the most predominant pattern. Also, imaging parameters of higher non-contrast CT value and higher SUVmax value are associated with non-predominant but high-grade histologic pattern.

**Abstract:**

Background: Prognostic considerations for non-predominant patterns are necessary because most lung adenocarcinomas (ADCs) have a mixed histologic pattern, and the spectrum of actual prognosis varies widely even among lung ADCs with the same most predominant pattern. We aimed to identify prognostic stratification by second most predominant pattern of lung ADC and to more accurately assess prognostic factors with CT imaging analysis, particularly enhancing non-predominant but high-grade pattern. Methods: In this prospective study, patients with early-stage lung ADC undergoing curative surgery underwent preoperative dual-energy CT (DECT) and positron emission tomography (PET)/CT. Histopathology of ADC, the most predominant and second most predominant histologic patterns, and preoperative imaging parameters were assessed and correlated with patient survival. Results: Among the 290 lung ADCs included in the study, 231 (79.7%) were mixed-pathologic pattern. When the most predominant histologic pattern was intermediate-grade, survival curves were significantly different among the three second most predominant subgroups (*p* = 0.004; low, lepidic; intermediate, acinar and papillary; high, micropapillary and solid). When the second most predominant pattern was high-grade, recurrence risk increased by 4.2-fold compared with the low-grade group (*p* = 0.005). To predict a non-predominant but high-grade pattern, the non-contrast CT value of tumor was meaningful with a lower HU value associated with the histologic combination of lower grade (low-grade as most predominant and intermediate-grade as second most predominant pattern, OR = 6.15, *p* = 0.005; intermediate-grade as most predominant and high-grade as second most predominant pattern, OR = 0.10, *p* = 0.033). SUVmax of the tumor was associated with the non-predominant but high-grade pattern, especially in the histologic combination of intermediate-high grade (OR = 1.14, *p* = 0.012). Conclusions: The second most predominant histologic pattern can stratify lung ADC patients according to prognosis. Thus, predicting the malignant potential and establishing treatment policies should not rely only on the most predominant pattern. Moreover, imaging parameters of non-contrast CT value and SUVmax could be useful in predicting a non-predominant but high-grade histologic pattern.

## 1. Introduction

Invasive lung adenocarcinoma (ADC) has been classified by the 2011 classification system of the International Association for the Study of Lung Cancer (IASLC), American Thoracic Society (ATS), and European Respiratory Society (ERS) into five distinct histological patterns: lepidic, acinar, papillary, micropapillary, and solid [1]. Lung ADC is divided into three prognostic groups according to the most predominant pattern detected by histopathology [2,3,4]. However, the main limitation of the predominant pattern concept is that most lung ADCs are mixed-patterns, and only 6% to 22% of ADCs are pure-pathologic patterns that consist of a single histologic pattern [5,6]. Among lung ADCs with the same most predominant pattern, the spectrum of actual prognosis varies widely, especially for intermediate-grade lung ADCs [3,7,8,9]. Thus, prognostic considerations for non-predominant histologic patterns of lung ADCs are needed.

According to a paper recently published by the IASLC pathology group, the predominant plus high-grade pattern classified patient prognosis is better than the predominant pattern alone. This result is consistent with the results reported by Takahashi et al. that the presence of a very small portion of the high-grade pattern deteriorates patient prognosis [8]. The IASLC pathology group also reported that the combination of the two most predominant patterns has similar prognostic distinguishing ability to the predominant plus high-grade pattern [10]. This suggests that it is important to consider the non-predominant histologic patterns, such as second most predominant pattern or small portion of high-grade pattern, when predicting patient prognosis. 

Moreover, 80% of lung cancer patients are clinically inoperable [11]. In these patients, the biopsy samples performed prior to chemotherapy show only a very small portion of the tumor and do not reflect the entire tumor. From this perspective, the need for non-invasive histologic prediction using imaging to predict not only the most predominant pattern, but also the non-predominant pattern is increasing.

Ito et al. showed a difference in recurrence-free survival depending on whether the second most predominant pattern was lepidic [7]. ADCs with a non-lepidic second most predominant pattern showed a higher rate of recurrence when the most predominant pattern was intermediate-grade. However, the authors took a dichotomous approach in examining the second most predominant pattern and categorizing cases as lepidic or not, where intermediate and high-grade pattern were dealt with as one subgroup.

Our study aimed to identify prognostic stratification by second most predominant histologic pattern of lung ADC when most predominant pattern is intermediate grade. In addition, a goal was to more accurately assess prognostic factor with CT imaging analysis, particularly enhancing non-predominant but high-grade pattern. To the best of our knowledge, this is the first study to attempt CT and prognostic analysis focusing on non-predominant patterns.

## 2. Material and Methods

This single-center prospective study was approved by the institutional review board of our institution (Samsung Medical Center), and informed consent was obtained from all patients.

### 2.1. Study Population

This study was conducted as a part of an ongoing prospective clinical trial of early stage lung ADC patients who underwent pre-operative DECT scans (NCT01482585) from December 2011 to January 2017. We included patients who were scheduled to undergo curative surgery. Patients with a history of previous radiation or chemotherapy were excluded.

### 2.2. Imaging and Analysis

CT images were obtained using a dual-source CT system (Somatom Definition Flash; Siemens Healthcare, Forchheim, Germany) with the dual-energy technique. Scanning was performed 90 sec after contrast media administration (100 mL of Iomeron 300; Bracco, Milan, Italy) at a rate of 1.5 mL/sec, followed by a 20 cc saline flush with same rate. 

With dual energy CT technology, which uses two X-ray tubes with different peak kilovoltages (kVp), spectral imaging and material decomposition/quantification are possible. Using this, discrimination of iodized blood in contrast-enhanced CT and quantification of iodine could be performed. Virtual non-contrast (VNC) image is an image post-processing technique used to generate non-contrast images of contrast-enhanced scans via subtraction of iodine [12]. This method has the advantage of reducing the radiation dose in that it is not necessary to obtain additional precontrast images when taking conventional contrast-enhanced CT.

The non-contrast and iodine-enhanced images were generated from altering the liver VNC application mode of the dedicated dual-energy post-processing software (Syngo Dual Energy; Siemens Medical Solutions, Forchheim, Germany). The Hounsfield unit (HU) value of fat in the liver VNC application mode was substituted with that of air, because the ground-glass opacity nodules are a mixture of air and soft tissue. Therefore, the material parameters were −110 HU for air at 80 kV, −115 HU for air at 140 kV, 60 HU for soft tissue at 80 kV, and 54 HU for soft tissue at 140 kV [13]. All images were reconstructed with a section thickness of 1 mm using a D30f (medium smooth) kernel for the iodine map and a D45f (medium sharp) kernel for the non-contrast images.

For nodule segmentation, an ROI was drawn with a semiautomatic approach on the axial images to generate a volume of interest. Additional manual correction was performed to exclude bronchovascular structures and the border of ground-glass opacity. Next, intensity value pairs (a pair consists of VNC and iodine values) were collected from the ROI. A joint histogram was constructed from the intensity pairs and reflects the distribution of intensity pairs (Figure 1). This might contain richer information than a single histogram in which only the intensity distribution of one modality is considered. Thus, the joint histogram could serve as a source for a distinct pattern for each patient. Additionally, we calculated the point of highest incidence of intensity pairs from the joint histogram (cross-point value) (Figure 1). Furthermore, we examined the localized area that contains 30% of the total distribution around the cross-point value.

### 2.3. PET Analysis

FDG PET/CT images were acquired using a PET/CT device (Discovery LS; GE Healthcare, Milwaukee, WI, USA), which consisted of a PET scanner (Advance NXi; GE Healthcare) and an eight-slice CT scanner (Light-Speed Plus; GE Healthcare). A nuclear medicine physician who was unaware of clinicopathologic information interpreted the PET images. ROIs were drawn over the most intense area of FDG uptake for semi-quantitative analysis. When it was not possible to evaluate nodular FDG uptake, an ROI was drawn in a presumed nodular location considering CT component images of PET/CT. FDG uptake within the ROIs was calculated as SUVmax.

### 2.4. Visual Assessment of Joint Histogram Graphs

We analyzed the shape and distribution of the joint histogram by visual assessment. In terms of shape, we divided the results into two groups: diffuse and taller than wide. “Taller than wide” was defined when the longitudinal length of the scatter plot was greater than twice the horizontal length (Figure 1B,C). Otherwise, it was defined as “diffuse” (Figure 1A,D). In terms of distribution, there were three groups: compact (Figure 1A,B), scattered (Figure 1C), and mixed (Figure 1D). The compact distribution is a collection of pixels without space between them, whereas the scattered distribution has space between the pixels. The mixed distribution is a mixture of two patterns, mainly a compact distribution and a scattered distribution similar to a comet tail. We also analyzed graphs that emphasized areas that contain 30% of the total distribution (area 30) and divided the results into two groups: compact (Figure 1A,D) and diffuse (Figure 1B,C). Compact area 30 means that all pixels are within 40 HU range in the iodine map. Lastly, according to VNC value of the cross-point, we generated three VNC groups: high (>−64, Figure 1A,D), intermediate (−544 to −64, Figure 1B), and low (<−544, Figure 1C). The cutoff of the VNC group was determined by referring to the median VNC values according to most predominant pattern and the interval empirically divided in the visual assessment. Mean CT numbers of iodine-enhanced images (mean iodine) were obtained. SUVmax obtained from PET/CT conducted before surgery also was analyzed.

In total, we evaluated imaging parameters from the joint histogram graph (shape, distribution, area 30, VNC group), the mean iodine value from iodine-enhanced images, and a functional imaging parameter (SUVmax) from PET/CT.

Additionally, the percentage of ground-glass opacity (GGO percentage) in the entire tumor was visually assessed. The GGO percentage was divided into four categories as follows: 100% (pure GGO), greater than or equal to 50%, greater than or equal to 25%, and less than 25% [14].

All visual assessments were performed by two chest radiologists with 4 and 17 years of experience in thoracic CT interpretation, respectively; clinical and pathologic results were unknown. In the case of different results of visual assessment between the two observers, consensus was achieved through review and further discussion.

### 2.5. Pathologic Evaluation

For pathologic evaluation, whole tumor tissue sections were obtained and placed on a slide. One experienced lung pathologist (22 years of experience in lung pathology) interpreted all tissue sections by virtual slides using ImageScope viewing software (12.3.3, Aperio Technologies, Inc., Seoul, South Korea) and a high-resolution monitor [14]. The second pathologist also reviewed the entire slides. In cases of disagreement of the histological pattern, consensus was reached after sufficient discussion. The evaluations were performed according to IASLC/ATS/ERS classification criteria, quantifying the extent of each histologic component [15]. Comprehensive histologic patterning was performed in a semi-quantitative manner to the nearest 5%, summing to a total of 100% pattern components per tumor. The most predominant pattern in a mixed-type tumor was determined by the histopathologic pattern that composed the greatest percentage of the tumor [14]. When evaluating the predominant histologic pattern, area of internal scar tissue was disregarded.

### 2.6. Statistical Analysis

Continuous and categorical variables are summarized as mean (standard deviation) and frequency (percentage), respectively. To identify independent factors for prediction of specific histologic combination of the most and second most predominant patterns, multiple logistic regression analyses were performed. For some of patients with several lung ADCs, we performed clustered logistic regression to estimate robust standard error. Variable visual assessment parameters from joint histogram and SUVmax value were evaluated. Ten-fold cross validation was performed for results of multiple logistic regression to assess internal validity.

Disease-free survival (DFS) was defined as the time interval from surgical resection to tumor recurrence, whether or not the disease-free state was assessed based on the last follow-up. DFS curves were plotted by the Kaplan–Meier method, and the curves of subgroups were compared using the log-rank test. Associations between pathologic subgroup and DFS were evaluated using a Cox proportional hazards model. Further, Kruskal–Wallis test was performed for identifying differentiation of SUVmax according to VNC group and histologic grade. Statistical significance was defined as two-tailed *p* < 0.05. Testing for multiple factors was corrected using Bonferroni’s method due to inflated type I error. Statistical analysis was executed using SAS version 9.4 (SAS Institute, Cary, NC, USA) and R 4.0.1 (Vienna, Austria; http://www.R-project.org, accessed on 31 December 2020).

## 3. Results

### 3.1. Patient Demographics

A total of 275 patients with 287 invasive lung ADC and three adenocarcinoma in situ (AIS) lesions was enrolled. The clinicopathologic characteristics are listed in Appendix A. Regarding the most predominant histologic subtype, 183 tumors (63.1%) was acinar pattern, followed by lepidic (48 tumors; 16.6%), papillary (31 tumors; 10.7%), solid (20 tumors; 6.9%), and micropapillary pattern (8 tumors; 2.7%). In terms of histologic grading of the most-predominant pattern, 48 (16.6%) tumors belonged to low-grade, 214 (73.8%) to intermediate-grade, and 28 (9.6%) to high-grade.

The relationships between most predominant and second most predominant histologic patterns are shown in Table 1. There were 59 (20.3%) tumors with pure-pathologic pattern and 231 (79.7%) tumors with mixed-pathologic pattern. Of the 231 tumors with mixed-pattern, 189 (81.8%), 38 (16.5%), and 4 (1.7%) tumors were composed of two, three, and four histologic patterns, respectively.

Of the 257 patients, 31 (12.1%) experienced recurrence. The mean DFS was 49.9 months, and the mean follow-up period was 52.6 months (range, 1–92 months). Eleven patients died during follow-up, and six of these patients died from recurrent lung cancer.

### 3.2. Prognostic Factors of Disease-Free Survival

We performed univariate and multivariate Cox regression analyses for predicting DFS. In the univariate analysis, shape, area 30, VNC group, second most predominant pattern, GGO percent, SUVmax, and pathologic stage were possible predictive factors for DFS (Appendix A). Among them, VNC group, second most predominant pattern, as well as pathologic stage were used for multivariate analysis. Multivariate analysis demonstrated that higher VNC group rating had higher risk of recurrence; the respective HRs (95% CI) were 0.517 (95% CI: 0.225–1.189), and 0.231 (95% CI: 0.073–0.733) for intermediate VNC group and low VNC group versus high VNC group. Pathologic stage remained a significant predictor for DFS. Higher grade pathologic stages have 1.7-fold higher risk of disease recurrence (95% CI: 1.358–2.047). We additionally performed multivariate analysis addressing SUV, histology, and GGO percent. Multivariate analysis revealed that higher VNC group rating had higher risk of recurrence; the respective HRs (95% CI) were 0.458 (95% CI: 0.158–1.327) and 0.197 (95% CI: 0.047–0.827) for intermediate VNC group and low VNC group versus high VNC group. The significance of other variables was lost when analyzed with the VNC group.

### 3.3. Prognosis Based on Second Most Predominant Pattern

The eight histologic combinations of most and second most predominant patterns showed significantly different DFS rates (*p* < 0.04) (Appendix A). There was no statistically significant difference of disease-free survival (DFS) according to the most predominant histologic pattern, especially between intermediate and high grade. Detailed 3- and 5-year disease-free survival rates for each of the most predominant histologic patterns are shown in Appendix A. We performed additional analysis of DFS stratified by the second most predominant pattern in cases with the most predominant pattern of intermediate-grade. Figure 2 demonstrates Kaplan–Meier survival curves for DFS based on second most predominant pattern. Survival curves were significantly different among the three second most predominant subgroups (*p* = 0.004). In particular, when the second most predominant group was high-grade, the risk of recurrence increased by 4.2-fold compared with the low-grade group (*p* = 0.005). Because of the heterogeneity of pathologic stage, rank test under complex sampling was performed for pathologic T and N stage by second most predominant group. Pathologic T stage was not significantly different among second most predominant groups when most predominant group was intermediate grade (*p* = 0.075). Pathologic N stage, on the other hand, tended to have a higher frequency of N-positive results when the second most predominant group grade was high (*p* < 0.001). Of 214 patients with intermediate grade in the most predominant group, 20 showed pathologic N1 or N2 stage. Interestingly, of 20 N-positive patients, 14 (70%) showed postoperative unexpected upstaging results compared to clinical N stage, as shown in Appendix A.

### 3.4. Prediction for Non-Predominant But Higher Second Most Predominant Pattern

We identified the predictors of clinically important histologic combinations for which the second most predominant pattern is higher grade than the most predominant pattern (most-second most predominant subgroup, low-intermediate grade and intermediate-high grade). 

Simple logistic regression analysis revealed that shape, iodine, VNC group, and SUVmax were statistically significant predictors of the low-intermediate grade histologic combination (Table 2). However, multiple logistic regression analysis demonstrated that only the VNC group was a significant predictor of low-intermediate grade. Low value of VNC showed a 6.2-fold higher probability of low-intermediate grade histologic combination (OR = 6.15, *p* = 0.005). 

For prediction of the intermediate-high grade histologic combination, imaging markers such as shape, distribution, VNC group, and SUVmax were statistically significant in simple regression analysis (Table 2). Multiple logistic regression analysis showed that VNC group and SUVmax were significant predictors of intermediate-high grade. Low value of VNC demonstrated a lower probability of intermediate-high histologic combination (OR = 0.10, *p* = 0.033). Higher metabolic activity (SUVmax) appeared to be associated with higher probability of intermediate-high grade (OR = 1.14, *p* = 0.017). A 10-fold cross validation (CV) method was used to examine the internal validity, and the variables that were meaningful in multiple regression analysis were important; the AUC using CV was 0.70 (95% CI; 0.597–0.797). 

### 3.5. Associations among SUVmax, VNC Group, and Histologic Grade

SUVmax was significantly different among the three VNC groups (*p* < 0.0001), which shows that tumors with high VNC values are associated with high SUVmax (Figure 3).

The three most-predominant histologic patterns showed significantly different SUVmax values (*p* < 0.0001): 1.4 ± 2.2 (mean ± SD) for low-grade, 2.9 ± 3.5 for intermediate-grade, and 6.0 ± 3.7 for high-grade. When the most predominant group was intermediate-grade, the high-grade second most predominant group showed significantly higher SUVmax than the low- and intermediate-grade groups (*p* < 0.0001, Appendix A). However, there was no significant difference between SUVmax of the low- and intermediate-grade second most predominant patterns.

The interrelationships between VNC, iodine map, SUVmax, and histological classification are demonstrated in Figure 4 and Appendix A.

## 4. Discussion

Since introduction of the 2011 classification system of IASLC/ATS/ERS, there has been increased interest in the associations between quality and quantity of histologic patterns of lung adenocarcinoma and patient prognosis [5,16,17,18,19]. Several studies have suggested that consideration of minor patterns in addition to the most predominant pattern results in better patient classification [5,7,20]. According to a paper recently published by the IASLC pathology group, the most predominant plus high-grade pattern or the combination of two most predominant patterns better classified patient prognosis than the most predominant pattern alone [10]. Thus, the current study examined the prognosis according to non-predominant histologic pattern of lung ADC, and our results showed that prognosis was stratified by second most predominant patterns.

In our study, 80% of tumors were mixed-pathologic patterns consisting of two or more histologic patterns, which is similar to the results of previous studies [6,21]. These results reaffirm that the single most predominant histologic pattern does not represent the entire tumor. In particular, acinar and papillary-predominant tumors, which are classified in the intermediate-grade group, have been reported to be highly heterogeneous with regard to tumor behavior compared with tumors of other patterns [3,22]. In addition, intermediate-grade tumors constitute a major proportion of ADCs, with approximately 50–70% of ADC being diagnosed as either acinar or papillary predominant pattern, irrespective of stage [2,4,17,19]. Therefore, we focused on this intermediate-grade of ADC, which accounts for most of the predominant pattern but exhibits varying tumor aggressiveness. 

Among cases in which the most predominant pattern was intermediate, DFS curves (Figure 2) clearly revealed a difference in prognosis according to second most predominant pattern of tumors (*p* = 0.004). The high-grade second most predominant group showed a 4.2-fold higher risk of recurrence compared with the low-grade group (OR = 4.22, *p* = 0.005). Ito et al. reported similar results showing prognostic differences according to non predominant patterns [7]. However, the second most predominant pattern was divided into low and intermediate to high grade. Our study is meaningful in that it was first to show a clear difference of prognosis according to grade of the second most predominant pattern. In particular, for patients with inoperable lung cancer, which account for about 80% of all lung cancer cases, the prognosis can vary depending on second most predominant pattern, even if the most predominant pattern is the same. The difference in pathologic N stages among the second most predominant patterns might have decreased the statistical significance of our results because the frequency of N-positive results tended to be higher when the second most predominant group grade was high. These results are consistent with reports that high grade lung cancer exhibits N-positive behavior [23]. Interestingly, however, a substantial portion of pathologic N-positive patients (14 out of 20, 70%) had unexpected postoperative upstaging results compared to clinical N stage, as shown in Appendix A. Given the recent trend toward limited pulmonary resection [24,25], this might be meaningful because unexpected N-positive result increase with high second most predominant group, which could allow us to find more accurate candidates for active LN dissection even at the recent circumferences of VATS. The study is meaningful because of possible correlation of second most predominant pattern with N-positive behavior.

The eight histologic combinations showed significantly different DFS curves (Appendix A). However, there was overlap of survival curves between the high-intermediate and intermediate-intermediate groups and between the high-intermediate and intermediate-high groups, reflecting that it is not sufficient to consider only the most predominant pattern. Further, there was an intersection of survival curves between the high-intermediate and high-high groups, which can be explained in part by the small sample size of high-grade tumors. Of the eight tumors with high-high grade, six (75%) were pure-solid pattern, while two (25%) consisted of solid as the most predominant pattern and micropapillary as the second most predominant pattern. The most predominant pattern in our study did not show statistical differences for DFS, especially between intermediate and high grades. Von der Thusen et al. reported that the most predominant micropapillary pattern did not show prognostic significance, partly because small numbers and relatively late stage of presentation [26]. Our study also demonstrated similar results, probably because of smallest number of micropapillary patterns and involving stage I to III patients. The fact that patients with micropapillary patterns suffered more follow-up loss may be another factor that prevents micropapillary patterns from having prognostic significance. However, when considering the second most predominant pattern as well as the most predominant pattern, the patients were stratified according to DFS. These results reaffirm that the second most predominant histologic pattern is related to the patient prognosis. 

In the clinical setting, intratumoral heterogeneity can lead to worse outcomes than expected for the patient, especially when the second most predominant pattern is a higher grade than the most predominant pattern, where more active treatment might be more effective. We searched for predictors of histologic combinations in which the second most predominant pattern had a higher grade. We found that VNC group was an independent predictor of both histologic combinations of low-intermediate and intermediate-high grades. However, the higher was the VNC group rating, the higher was the probability of being in the intermediate-high grade group, whereas the low-intermediate grade group had a lower VNC group rating. These results indicate that the higher is the VNC value reflecting the cellularity of the tumor, the higher is the histological grade [27]. Analysis with non-contrast CT imaging metrics could support quantification of tumor heterogeneity by evaluating the grayscale and predicting tumor cellularity [28,29]. Ikeda et al. conducted a similar study using non-contrast CT to distinguish ground-glass opacity nodules [29]. They showed that the 75th percentile CT number is optimal for differentiating AAH and BAC, and the mean CT number of the tumor can differentiate between BAC and adenocarcinoma. Our study, however, focused on early-stage ADC and revealed that VNC value is associated with histologic grade of ADC. 

The degree of tumor enhancement, as indicated by the iodine map, was not a significant predictor for a specific histologic combination, suggesting variable tumor enhancement. Many previous studies have reported that dynamic enhancement of nodules on CT can help differentiate malignant from benign nodules [30,31,32]. Nevertheless, in our study, it seems this does not apply with the malignant nodule, which can show a higher degree of contrast enhancement than in benign nodules, but they seem to contain various microvessel densities. In addition, variety of mucin production particularly in ADC can be another key factor of discrepant enhancement degree according to pathologic grade. 

SUVmax was another independent predictor of intermediate-high grade histologic combination. In our study, higher SUVmax was associated with higher probability of intermediate-high grade group (OR = 1.14, *p* = 0.017). FDG-PET SUVmax reflects the metabolic activity of tumors and is associated with prognosis and histology in lung cancer [33,34,35,36]. Lee et al. [13] reported that SUVmax allowed distinction of high-grade ADCs from other histologic grade tumors. Kadota et al. [37] reported that tumors with high-grade histology had a higher SUVmax. Our study also found that SUVmax significantly stratifies the three most predominant histologic patterns (*p* < 0.0001). Kadotal et al. conducted a sub-analysis of 18 patients with intermediate-grade histology but high SUVmax and reported that the patients had either high-grade second most predominant histologic pattern or high mitotic count. Similarly, in our study, SUVmax was useful in differentiating second most predominant patterns when the most predominant pattern was intermediate, suggesting that metabolic features reflect intratumoral heterogeneity. Several reports demonstrated positive correlation between FDG uptake and cellularity of tumors [38,39]. Figure 3 shows the differences of SUVmax among the three VNC groups (*p* < 0.0001). The VNC group reflects the degree of tumor cellularity. Therefore, our results are similar to those of previous studies, which means that tumors with high cellularity have high SUVmax.

When taking into account tumor characteristics, cellularity, perfusion, and metabolism of the tumor should be considered. Our study is meaningful in that we attempted to analyze all functional imaging parameters that reflect tumor biologic properties such as the values of VNC, iodine, and SUVmax, which reflect cell density, perfusion, and tumor metabolism, respectively. Further research with a larger patient group is necessary to predict precise intra-tumoral heterogeneity using these quantitative imaging parameters. In our study, high non-contrast CT values or high SUVmax values could predict histologic combination of a non-predominant but high grade pattern. By considering the second most predominant pattern as well as most predominant pattern, our results can be utilized to enable personalized treatment, reflecting the characteristics of highly heterogeneous lung ADC as much as possible. There are several reports that more active adjuvant chemotherapy has benefits for DFS in lung ADC patients with high-grade most predominant histologic pattern [40,41]. Similarly, adjuvant chemotherapy can be considered after sufficient multidisciplinary discussion, when surgical pathology shows a higher grade non-predominant histologic pattern in patients with early-stage operable lung cancer. On the other hand, in patients with advanced inoperable lung cancer, pre-chemotherapeutic biopsy sample cannot sufficiently predict histologic pattern of the entire tumor. In this case, non-contrast CT values or SUVmax values can indirectly imply a combination of high grade histologic pattern.

Our study has several limitations. First, this study was performed in a single institution. Second, although as many cases as possible were collected in this prospective study, there was an imbalance in the distribution of histological grades. Most cases were of intermediate-grade, with relatively few low- or high-grade histologic patterns. Thus, analysis of prognostic stratification by second most predominant pattern in these low- or high-grade tumors was difficult. However, analysis of intermediate histologic grade makes sense because it actually constitutes the largest proportion of the histologic grade in the global population [2,4,17]. To determine the impact of the second most predominant pattern in low- or high-grade tumors, further studies are needed with a larger number of patients. Third, external validation using an independent population was not conducted. However, we performed 10-fold cross-validation as a method of internal validation.

In conclusion, the histologic second most predominant pattern can stratify lung ADC patients according to prognosis when the most predominant pattern is intermediate-grade. Thus, predicting the malignant potential and establishing treatment policies should not rely only on the most predominant pattern. In addition, values of VNC and SUVmax could be useful in predicting non-predominant but higher-grade pattern of the tumor. These imaging parameters provide practical help in planning patient treatment in clinical practice.

## Figures and Tables

**Figure 1 cancers-13-02785-f001:**
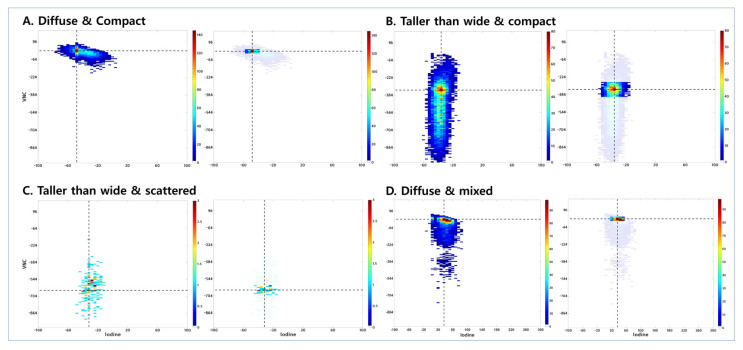
Joint histogram of VNC and iodine map. (**A**) A 62-year-old male patient with solid predominant type invasive adenocarcinoma. The joint histogram graph is shown on the left with the transverse axis as iodine histogram attenuation scale and the longitudinal axis as VNC histogram scale. The shape of intensity value pairs is diffuse, and the distribution is compact. The VNC group is within the high group. On the right, the graph shows the localized area that contains 30% of the total distribution around the cross-point value (area 30). Area 30 is compact. (**B**) A 62-year-old male patient with pure-acinar type invasive adenocarcinoma. The shape of the intensity value pairs is taller than wide, and the distribution is compact. The VNC group is within the intermediate group, and area 30 is diffuse. (**C**) A 62-year-old female patient with pure-lepidic type adenocarcinoma in situ (AIS). The shape of intensity value pairs is taller than wide, and the distribution is scattered. The VNC group is within the low group and area 30 is diffuse. (**D**) A 46-year-old male with papillary-predominant type invasive adenocarcinoma. The shape of intensity value pairs is diffuse, and the distribution is mixed. The VNC group is within the high group, and area 30 is compact.

**Figure 2 cancers-13-02785-f002:**
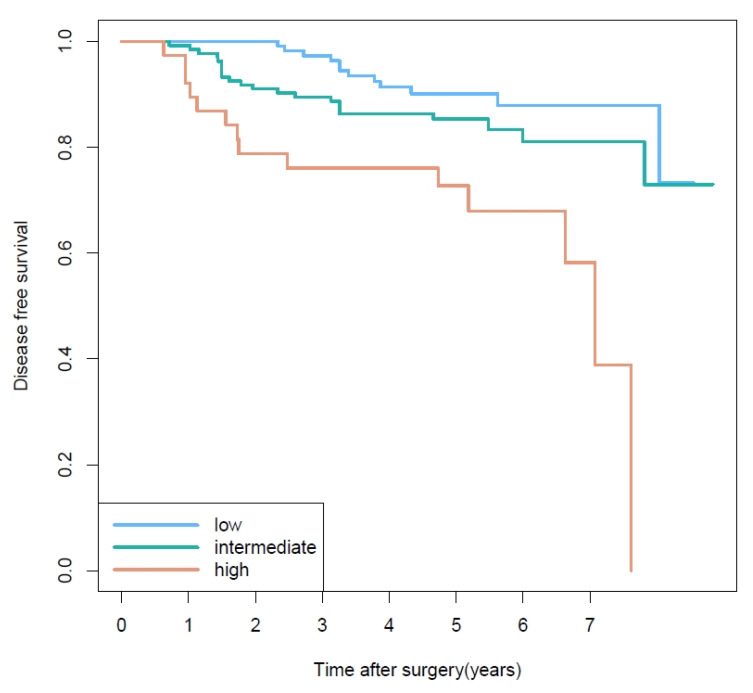
Disease-free survival curves based on the second most predominant pattern when the most predominant pattern is intermediate grade. Survival curves were significantly different among the three second most predominant subgroups (*p* = 0.004).

**Figure 3 cancers-13-02785-f003:**
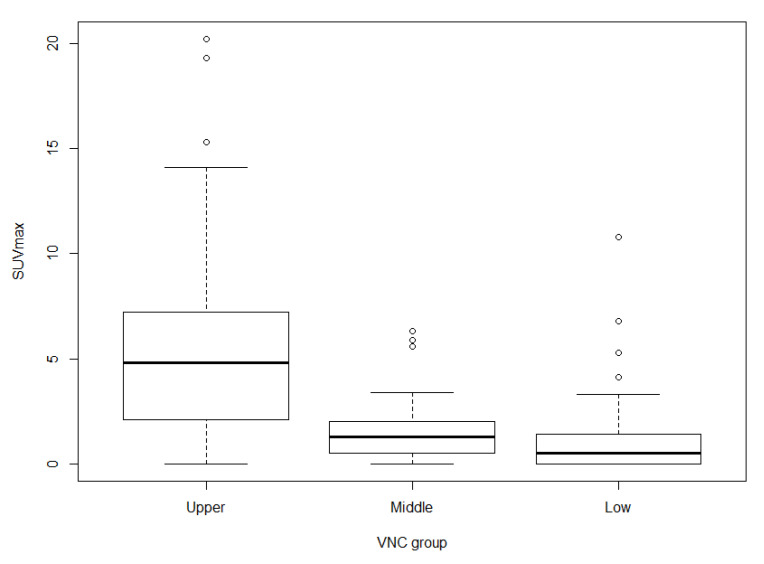
Distribution of SUVmax for each VNC group. SUVmax of lung adenocarcinomas categorized according to VNC group (*p* < 0.0001). The boxes extend from the 25th percentile to the 75th percentile, and the whiskers extend to 1.5 times the interquartile distance.

**Figure 4 cancers-13-02785-f004:**
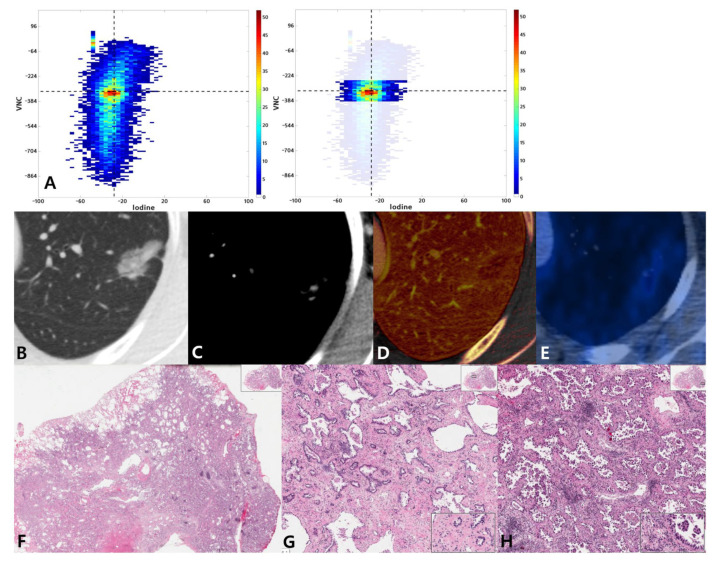
A 54-year-old woman with lung adenocarcinoma with a 70% acinar and 30% micropapillary pattern. (**A**) Joint histogram showing a taller than wide shape and compact distribution pattern. (**B**) Targeted view of the lung window VNC image shows a 24-mm-sized part-solid nodule in the left upper lobe. (**C**) On conventional enhanced image with mediastinal window, tiny areas of solid portion are seen. (**D**) On the iodine map, the range of mean iodine value was −51 to 136. (**E**) PET/CT image shows FDG uptake with SUVmax of 1.6. (**F**) Photomicrograph (hematoxylin and eosin stain, 10×) shows invasive adenocarcinoma with acinar predominant pattern. High magnification (HE, 100×) shows acinar pattern (**G**) and micropapillary pattern (**H**) lung adenocarcinoma.

**Table 1 cancers-13-02785-t001:** Relationships between most predominant and second most predominant histologic patterns.

Most Predominant Pattern	No. of Lesions with Second Most Predominant Pattern					Total No. (%)
	Lepidic	Acinar	Papillary	MP	Solid	
Lepidic	3 ^†^	45	0	0	0	48 (16.6)
Acinar	90	44 ^†^	31	6	12	183 (63.1)
Papillary	2	15	6 ^†^	4	4	31 (10.7)
MP	1	4	3	0 ^†^	0	8 (2.7)
Solid	1	11	0	2	6 ^†^	20 (6.9)
Total No. (%)	97 (33.5)	119 (41.0)	40 (13.8)	12 (4.1)	22 (7.6)	290 (100)

Note. MP = micropapillary, ^†^ = number of lesions with pure pathologic pattern. Histologic subgroups were divided into low grade (lepidic), intermediate grade (acinar and papillary), and high grade (micropapillary and solid) of prognostic significance.

**Table 2 cancers-13-02785-t002:** Predictors of a histologic combination with non-predominant but higher grade pattern.

Imaging Parameters	Univariate			Multivariate		
	OR	95% CI	*p* Value	OR	95% CI	*p* Value
**Low grade as the most-predominant and intermediate grade as the second most predominant pattern**						
Shape	2.28	1.05–4.97	**0.038**	1.58	0.65–3.85	0.316
Distribution (scattered)	1.97	0.87–4.44	0.103	NA		
Distribution (mixed)	0.53	0.23–1.24	0.146	NA		
Distribution (compact)	Ref			NA		
Area 30	1.70	0.75–3.83	0.205	NA		
VNC group (middle)	2.17	0.76–6.21	0.148	NA		
VNC group (low)	10.29	3.99–26.49	**<0.0001**	6.15	1.72–21.95	**0.005**
VNC group (upper)	Ref			NA		
Mean iodine	0.97	0.94–0.99	**0.007**	0.98	0.95–1.02	0.318
SUVmax	0.77	0.64–0.94	**0.008**	0.93	0.77–1.12	0.421
**Intermediate grade as the most-predominant and high grade as the second most predominant pattern**						
Shape	0.38	0.18–0.79	**0.010**	0.63	0.26–1.50	0.298
Distribution (scattered)	0.60	0.13–2.81	0.521	NA		
Distribution (mixed)	2.98	1.36–6.52	**0.006**	2.01	0.82–4.97	0.128
Distribution (compact)	Ref			NA		
Area 30	0.48	0.23–1.03	0.060	NA		
VNC group (middle)	0.25	0.09–0.63	**0.004**	0.47	0.16–1.38	0.170
VNC group (low)	0.05	0.01–0.38	**0.004**	0.10	0.01–0.83	**0.033**
VNC group (upper)	Ref			NA		
Mean iodine	1.02	0.99–1.04	0.221	NA		
SUVmax	1.22	1.12–1.34	**<0.0001**	1.14	1.02–1.27	**0.017**

Note. CI = confidence interval, NA = not available, Ref = reference. Bold font = *p* < 0.05.

## Data Availability

The data presented in this study are available on request from the corresponding author.

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
