# Peer review of "Rethinking a Non-Predominant Pattern in Invasive Lung Adenocarcinoma: Prognostic Dissection Focusing on a High-Grade Pattern"

_cancers, 2021, doi:10.3390/cancers13112785_

Round 1

Reviewer 1 Report

I enjoyed the paper - I would suggest to give more relevance to the prognostic stratification of the histotypes. Right now, the distinction about ADC histotypes having different prognostic outlook, ie, low, intermediate and high is included in the legend of table 1 and briefly hinted to in the text. I wonder whether the Table could be modified or maybe an additional table is necessary which would summarize the prognostic distribution of ADC subsets and the literature references justifying this distinction.

Author Response

Thanks for your helpful comment.

As your recommendation, we added Supplementary Table 3 about detailed 3- and 5-year disease-free survival (DFS) rates for each of the most predominant histologic patterns.

In our study, there was no statistically significant difference of DFS according to the most predominant histologic pattern, especially between intermediate and high grade. Von der Thusen et al. reported that the most predominant micropapillary pattern did not show prognostic significance, partly because small numbers and relatively late stage of presentation1. Our study also demonstrated similar results, probably because of smallest number of micropapillary patterns and involving stage I to III patients. The fact that patients with micropapillary patterns suffered more follow-up loss may be another factor that prevents micropapillary patterns from having prognostic significance. However, when considering the second most predominant pattern as well as the most predominant pattern, the patients were stratified according to DFS. These results reaffirm that the second most predominant histologic pattern is related to the patient prognosis.

We reflected this on Results and Discussion section.

Supplementary Table 3. Disease-free survival according to most predominant pattern

Most predominant pattern

No. at risk

event

3 year DFS

(95% CI)

5 year DFS

(95% CI)

Lepidic

48

3

0.979 (0.938-1.000)

0.957 (0.900-1.000)

Acinar

183

35

0.889 (0.844-0.936)

0.834 (0.781-0.892)

Papillary

31

4

0.931 (0.843-1.000)

0.887 (0.772-1.000)

Micropapillary

8

1

0.875 (0.673-1.000)

N/A

Solid

20

5

0.895 (0.767-1.000)

0.739 (0.540-1.000)

Note. – The log rank test could not reject the null hypothesis of the difference of DFS between five most predominant histologic patterns. CI = confidence interval, DFS = disease-free survival, N/A = not available

1von der Thüsen JH, Tham YS, Pattenden H, Rice A, Dusmet M, Lim E, Nicholson AG. Prognostic significance of predominant histologic pattern and nuclear grade in resected adenocarcinoma of the lung: potential parameters for a grading system. J Thorac Oncol. 2013 Jan;8(1):37-44. doi: 10.1097/JTO.0b013e318276274e. PMID: 23242436

Reviewer 2 Report

I believe the revised paper has been a lot improved in all the parts. references has also been updated with a very important  paper published in 2020 on Journal of Clinical Oncology by IASLC pathologu task force. The paper  is now ready for publication

Author Response

Thanks for your response.

The English language has been further reviewed and carefully revised. We have attached a certificate of proofreading in English.

Reviewer 3 Report

The authors have made most of the recommended changes to the manuscript.

Author Response

Thanks for your response.

This manuscript is a resubmission of an earlier submission. The following is a list of the peer review reports and author responses from that submission.

Round 1

Reviewer 1 Report

This is an interesting contribution to the knowledge of prognosticators for resected ADC. However, the paper seems confused and difficult to read. There are several issues to address prior to considering this paper for publication:

  1. The rationale for this paper emerges like the tip of an iceberg only in a few lines of the manuscript (257-259 and 346-350). The paper should be radically rewritten based on these clearly outlined statements. 
  2. The originality of the paper is also somehow hidden in lines 277-279; the statement made at lines 65-67 is insufficient to substantiate the reason why this paper should be published. If one reads only the abstract, the conclusion section does not support originality either. 
  3. The concept of VNC should be expanded and the role of subjectivity in the radiological assessment should be clarified - the years of seniority in practice may not be per se a guarantee of reduced interobserver variability. In addition, the authors seem to take for granted  that our Readership is familiar with radiological terminology but since the paper is based on the VNC value in predicting tumor grade, it is worth specifying the detail of this type of assessment.
  4. The authors should discuss on how the histologic pattern would have an independent prognostic value in the presence of a substantial heterogeneity of stages (ie, more than 20% T size greater than pT1; more than 10% pN+). In fact, I am not sure which factors were included in the multivariable analysis.
  5. What is the clinical impact of this study? Should imaging indicate high grade first predominant/second predominant tumors, would the authors support the use of induction treatment irrespective of staging? The pattern definition is sometimes preoperatively available from bioptic, non surgical, specimens - would the authors support routine FNAB before surgery? Would resected high grade first predominant/second predominant tumors warrant adjuvant therapy?
  6. The English language should be carefully amended.

Reviewer 2 Report

This is a Prospective study to identify prognostic severity Among 290 lung ADCs  considering the non predominant histologic pattern of ADC and to more accurately perform prognostic stratification with CT imaging analysis particularly enhancing the non-predominant but high-grade pattern.
The non-predominant histologic pattern can stratify patients according to prognosis.Imaging parameters of non-contrast CT value and SUVmax can predict a non-predominant but high-grade histologic pattern.The non-predominant histologic pattern can stratify patients according to prognosis.
Imaging parameters of non-contrast CT value and SUVmax can predict a non-predominant but high-grade histologic pattern. I suggest

1)to take into account histology and staging of the lesion for prognostic evaluation and to add a table.

2)to add a multivariate analysis addressing SUV, histology , GGO predominant 

Reviewer 3 Report

The authors (Choi et al.) have investigated to identify prognostic severity according to non-predominant pattern of invasive lung adenocarcinoma and to more accurately perform prognostic stratification with CT imaging analysis, particularly enhancing non-predominant but high-grade pattern.

Their study is interesting.  However, there are some weaknesses as described bellows.

  1. The most important question is whether survival curves are significantly different among the three most predominant subgroups (low, intermediate, and high as the most predominant pattern) or not. Did the author obtain result showing survival curves was the worst in high-grade group and the best in low-grade group?  Judging from the supplementary Figure 1, I doubt it, because in the figure disease free survival is the worse in intermediate-high group than high-high group.  In this situation, I suspect that the lung adenocarcinoma patients the authors collected is not adequate for analyzing the effect of the second most predominant histologic pattern on survival.  The authors should respond to this question.
  2. Pathologist should be included in the author members, because pathological diagnosis is very important in this paper. Moreover, the pathological slides should be checked by at least two pathologists.
  3. In Table 2, “low grade as the most-predominant & intermediate or high grade as the second most predominant pattern” should be included.
  4. The authors described the following sentence “When the most predominant group was intermediate-grade, the high-grade second most predominant group showed significantly higher SUVmax than the low- and intermediate- grade groups (p < .0001)” in lines 233-235. The authors should provide a graph corresponding to this description.
  5. In Figure 2, “predominent type” should be revised to “predominant type”.
  6. The authors generated three VNC groups: high (>-64), intermediate (-544 to -64) and low (<-544). Please explain why these values were used.
  7. In this study, invasive lung adenocarcinomas were collected. According to the Figure 1, three lung adenocarcinomas showing pure lepidic pattern was included in this study.  Please describe the reason why these three cases were categorized in invasive adenocarcinoma in supplementary material.
  8. In supplementary Figure 3F, the authors described histology of the panel is micropapillary pattern. But I feel it is not typical micropapillary pattern.  Please exchange the figure with typical one.